# Prediction of Small for Gestational Age and Growth-Restricted Neonates at 35 to 36 Weeks of Gestation: A Multicenter Cohort Study

**DOI:** 10.3390/medicina61091626

**Published:** 2025-09-08

**Authors:** Raquel Martin-Alonso, Catalina de Paco Matallana, Nuria Valiño, Petya Chaveeva, Themistoklis Dagklis, Antonios Siargkas, Alan Wright, Mario Camacho, Valeria Rolle, Belén Santacruz, Maria M. Gil

**Affiliations:** 1Department of Obstetrics and Gynecology, Hospital Universitario de Torrejón, 28850 Torrejón de Ardoz, Madrid, Spain; 2School of Medicine, Universidad Francisco de Vitoria, 28223 Pozuelo de Alarcón, Madrid, Spain; 3Spanish Network in Maternal, Neonatal, Child and Developmental Health Research (RICORS-SAMID, RD24/0013/0018), Instituto de Salud Carlos III, 28040 Madrid, Spain; 4Department of Obstetrics and Gynecology, Hospital Clínico Universitario ‘Virgen de la Arrixaca’, 30129 El Palmar, Murcia, Spain; 5Institute for Biomedical Research of Murcia, IMIB-Arrixaca, 30129 El Palmar, Murcia, Spain; 6Department of Obstetrics and Gynecology, Complejo Hospitalario Universitario A Coruña, 15006 A Coruña, Galicia, Spain; 7Fetal Medicine Unit, Shterev Hospital, 1330 Sofia, Bulgaria; 8Pleven Medical University, 5800 Pleven, Bulgaria; 9Third Department of Obstetrics and Gynecology, School of Medicine, Faculty of Health Sciences, Aristotle University of Thessaloniki, 54124 Thessaloniki, Greece; 10Institute of Health Research, University of Exeter, Exeter EX4 4QJ, UK; 11Facultad de Estudios Estadísticos, Universidad Complutense de Madrid, 28040 Madrid, Spain; 12Plataforma de Bioestadística y Epidemiología, Instituto de Investigación Sanitaria del Principado de Asturias, Avenida Hospital Universitario s/n, 33011 Oviedo, Spain; 13Statistics Department, Fundación para la Investigación y el Desarrollo de la Medicina Materno-Fetal y Neonatal, iMaterna, 28806 Alcalá de Henares, Madrid, Spain

**Keywords:** small for gestational age (SGA), fetal growth restriction (FGR), third-trimester ultrasound, Doppler, PlGF, sFlt-1, prediction model, screening, pregnancy, birthweight percentile

## Abstract

*Background and Objectives*: Third-trimester screening is widely used to identify small for gestational age (SGA) and fetal growth restriction (FGR), but optimal models and timing remain under investigation. This study aimed to assess the performance of combined maternal factors and biomarkers, including ultrasound estimated fetal weight (EFW), Doppler indices, mean arterial pressure (MAP), and angiogenic biomarkers, for predicting SGA neonates after a routine 35–36 weeks’ scan in an unselected population. *Materials and Methods:* We conducted a retrospective cohort study in three Spanish centers offering universal third-trimester ultrasound. Logistic regression analyses were carried out to predict birthweight < 10th and <5th percentile using maternal characteristics and medical history, EFW, MAP, Doppler indices, and the angiogenic biomarkers placental growth factor (PlGF) and soluble fms-like tyrosine kinase-1 (sFlt-1). Using a 10-fold cross-validation, we estimated the area under the receiver operating characteristic curve (AUC), detection rates (DRs), false-positive rates (FPRs), and their corresponding screen-positive rates (SPRs). External validation was performed using an independent cohort. *Results:* Among 3992 pregnancies, the DR of ultrasound alone for birthweight <10th percentile was 47.9% (95% CI: 44.0 to 51.9), with an FPR of 7.3%. Adding maternal factors increased DR to 57.0% (95% CI: 53.0 to 60.9) at 10% FPR and to 83.0% (95% CI: 79.9 to 85.9) at 30% FPR. Similarly, the DR of ultrasound alone for birthweight < 5th percentile was 48.4% (95% CI: 43.1 to 53.6), with an FPR of 4.5%. Adding maternal factors increased DR to 65.7 (95% CI: 60.5 to 70.5) at 10% FPR and to 88.2 (95% CI: 84.4 to 91.3) at 30% FPR. The inclusion of MAP, Doppler, and biomarkers provided marginal additional gains, particularly for <5th percentile prediction. To achieve a DR > 80%, an SPR of approximately 40% was required. Performance improved when focusing on neonates born before 38 weeks, with a DR of 77.5 (95% CI: 68.6 to 84.9) at 10% FPR for SGA < 10th percentile. However, less than 40% of screen-positive women remained undelivered by 40 weeks, limiting the number requiring further surveillance. *Conclusions*: A third-trimester screening at 35–36 weeks using maternal characteristics and EFW identifies most SGA neonates, particularly those delivering before 38 weeks. Even including other biomarkers, an SPR of about 40% should be necessary to achieve a high DR. However, less than 40% of the women would remain undelivered before a subsequent follow-up is required.

## 1. Introduction

Small-for-gestational-age fetuses, defined as those with a birthweight below the 10th percentile, and especially those with fetal growth restriction, the most severe side of the spectrum, are at increased risk of adverse perinatal outcomes, including perinatal morbidity, fetal demise, and neonatal death, particularly as pregnancy progresses beyond term [1,2]. In the longer term, these children also face higher risks of suboptimal neurodevelopment, as well as adult-onset cardiovascular and metabolic disorders such as obesity, type 2 diabetes, coronary heart disease, and stroke [3].

Over the past decades, several studies have aimed to improve the antenatal detection of SGA fetuses, as identifying these fetuses allows for closer monitoring, timely delivery, and appropriate neonatal care, strategies that have been shown to improve outcomes [1,4,5]. However, the diagnostic accuracy of third-trimester ultrasound remains suboptimal. The average error in fetal weight estimation is around 8%, rising to 13% at the extremes of fetal size, particularly those classified as SGA or large-for-gestational-age fetuses (LGA, birthweight > 90th percentile) [6,7]. Thus, a 2015 meta-analysis by Bricker et al., which included 34 studies, found no significant benefit of third-trimester ultrasound on perinatal outcomes [8]. Moreover, the indiscriminate use of late-pregnancy ultrasound has been considered by many authors a cause of unintended harm: With a false positive rate (FPR) for FGR estimated at 10–15%, many fetuses of normal weight are misclassified, leading to unnecessary monitoring, interventions, increased induction rates, overuse of healthcare resources, and heightened anxiety for expectant parents [9,10].

In contrast, other studies suggested that routine third-trimester ultrasound in low-risk pregnancies may improve detection of SGA and FGR, especially when performed between 34 + 0 and 36 + 6 weeks, a gestational window with the highest diagnostic yield for growth abnormalities [11,12]. Reflecting this, the International Society of Ultrasound in Obstetrics and Gynecology (ISUOG) has recommended, since 2019, third-trimester screening for FGR as a core component of prenatal care [13], and it is routinely practiced across Spain, as recommended by the Spanish Society of Obstetrics and Gynecology (SEGO) [14].

For years, it has been established that the optimal strategy for detecting FGR combines sonographic fetal weight estimation, most effectively using the Hadlock 3 formula [15,16], with maternal clinical characteristics and obstetric history [17]. Recent studies using these combined models have reported DR for SGA as high as 80% for neonates born within two weeks of ultrasound assessment, with an FPR of approximately 10%. However, when considering SGA infants born at any point after the scan, DRs decline to around 65% [18,19].

The aim of this study was to evaluate whether the predictive performance of routine third-trimester screening for SGA and FGR, both in pregnancies delivering before 38 weeks and at any gestational age following screening at 35 + 0 to 36 + 6 weeks can be improved by combining maternal clinical data with sonographic fetal biometry, Doppler parameters, and angiogenic biomarkers, including placental growth factor (PlGF) and soluble fms-like tyrosine kinase-1 (sFlt-1), or the sFlt-1/PlGF ratio.

## 2. Materials and Methods

### 2.1. Study Design and Population

This was a retrospective cohort study conducted between January 2017 and December 2019 at three fetal medicine units in Spain, where a routine third-trimester scan is performed at 35 + 0 to 36 + 6 weeks: Hospital Clínico Universitario Virgen de la Arrixaca, in Murcia, Hospital Universitario de Torrejón, in Madrid, and Complejo Hospitalario Universitario de A Coruña, in Galicia. We included all uncomplicated singleton pregnancies that attended a routine ultrasound appointment between 35 + 0 and 36 + 6 weeks of gestation at any of the participating centers. We excluded patients under 18 years of age, those pregnancies that lacked adequate first-trimester dating (defined as crown-rump length or biparietal diameter measurement at 11–13 weeks [20,21] or in the case of in vitro fertilization, dating based on the date of conception), pregnancies with current monitoring due to suspected fetal growth impairment, patients with serious mental health conditions or learning disabilities that impede consent, or if informed written consent was not obtained, and pregnancies lost to follow-up.

At the third-trimester visit, maternal characteristics and medical history related to adverse pregnancy outcome were recorded, including maternal age, height, weight, racial origin (White, Black, South-Asian, East-Asian, or mixed), smoking status (yes/no), parity (nulliparous or parous, defined as no previous delivery beyond 24 weeks), and method of conception (spontaneous or assisted). Additional clinical variables included the presence of systemic lupus erythematosus (SLE), antiphospholipid syndrome (APS), type 1 or 2 diabetes, chronic hypertension, family history of preeclampsia (PE), personal history of PE and/or FGR, current gestational diabetes, PE, and pregnancy-induced hypertension (PIH) [22].

Fetal weight was estimated during the 35 + 0–36 + 6 week ultrasound using standard biometric parameters, head circumference, abdominal circumference, and femur length, according to the Hadlock 3 formula [15]. All ultrasound examinations were performed by certified sonographers following standard protocols.

As part of a parallel prospective study [23], in Murcia and Madrid, measurement of the MAP was carried out with automated and validated devices (Microlife, Taipei, Taiwan) according to a standardized protocol [24]. Additionally, transabdominal Doppler studies were also performed to assess the pulsatility indices (PIs) of the uterine artery (UtA), umbilical artery (UA), and middle cerebral artery (MCA); and serum concentrations of PlGF and sFlt-1 [25] were measured between 35 and 36 weeks of gestation using an automated immunoassay platform (BRAHMS KRYPTOR compact PLUS, Thermo Fisher Scientific, Hennigsdorf, Germany), as part of a parallel prospective study [23]. These biomarker data were not available for the cohort from Galicia.

### 2.2. Outcome Measures

The outcome measures were delivery of an SGA neonate below the 10th or the 5th percentile, born after the third-trimester ultrasound and obstetric assessment. The Fetal Medicine Foundation (FMF) fetal and neonatal population weight charts were used to calculate percentiles and z-scores for birthweight [26]. Pregnancies’ outcomes were retrieved from hospital/regional records or by contacting the delivery hospitals or the women’s general medical practitioners/midwives.

### 2.3. Statistical Analysis

Descriptive data were expressed as median and interquartile range (IQR) for continuous variables and as absolute and relative frequencies for categorical variables. Group comparisons were performed using the Mann–Whitney U-test or Fisher’s exact test as appropriate. UtA-PI, UA-PI, MCA-PI, PlGF, and sFlt-1 were converted into MoMs to adjust for maternal and pregnancy characteristics using the FMF online calculator [27].

A series of logistic regression models was constructed to assess the ability to predict neonates born with birthweight below the 10th and 5th percentiles at any gestational age after ultrasound assessment. Variables were selected based on clinical relevance. The baseline models for each outcome included gestational age at ultrasound, EFW z-score, maternal age, height, weight, smoking status, mode of conception, presence of SLE, APS, diabetes, chronic hypertension, family history of PE, personal history of PE and/or fetal growth restriction, current gestational diabetes, PE, and PIH. A second set of models additionally included MAP MoMs. A third set incorporated Doppler indices (UA-PI, MCA-PI, and UtA-PI MoMs), and a final set included biomarkers (PlGF and sFlt-1 MoMs) in addition to the above variables. A spline function for maternal age was initially tested to account for potential non-linear associations, but it was excluded due to poorer model performance.

Model performance was assessed by evaluating both discrimination and calibration. Discrimination was measured as the model’s ability to distinguish between SGA and non-SGA cases, estimating the DR at different fixed FPRs of 10%, 20%, 30%, and 40%. Ten-fold cross-validation was used to estimate predictive performance. The dataset was randomly divided into ten subsets; in each iteration, nine subsets were used for model training and one for validation, with this process repeated until all subsets had served as the validation fold. The area under the receiver operating characteristic (ROC) curve (AUC), DR at each FPR, and corresponding 95% CI were calculated from the pooled predictions across all folds. Finally, SPRs corresponding to each FPR were also calculated.

Calibration was evaluated by comparing predicted risks against observed outcomes, i.e., the proportion of neonates correctly identified as having birthweight < 10th or <5th percentiles. In a perfect calibration, the slope should be 1 and the intercept should be 0.

External validation was subsequently performed using an independent dataset from Hospital Universitario de A Coruña (Galicia). As not all variables were available in this cohort, a reduced model was constructed using the same methodology but limited to the available variables.

Since in clinical practice, fetuses with an EFW < 10th percentile are typically followed up after two weeks or recommended for delivery at 37 weeks if classified as FGR, we also planned to evaluate the model’s predictive performance in the subgroup of patients who delivered before 38 weeks and the percentage of women who would require an additional scan.

All analyses were conducted on a complete case basis using R version 4.5.0, with the ROCR package for ROC curves and epiR for DR calculations [28,29]. A *p*-value < 0.05 was considered statistically significant.

## 3. Results

### 3.1. Patient Characteristics

A total of 3992 women in Murcia and Madrid had all the necessary data for the development of the model, and 551 women from Galicia were used as external validation. Demographic and pregnancy characteristics according to SGA status are shown in Table 1, and differences between the development and validation cohorts are shown in Table 2. Women who delivered SGA neonates were older, shorter, and had lower maternal weight compared to those who delivered non-SGA neonates. There was a higher prevalence of smoking, as well as a greater proportion of women with a current diagnosis of PE or PIH, or a history of previous FGR, among mothers of SGA neonates. In terms of biomarkers, these women had higher MAP, UA-PI and UtA-PI, and sFlt-1 levels, alongside lower MCA-PI and PlGF levels. Significant differences were also observed between the two populations. Women in the Galicia cohort were, on average, older, shorter, and had lower maternal weight compared to those from Murcia and Madrid. They also had a higher prevalence of chronic hypertension and diabetes, and a greater proportion were nulliparous or conceived through assisted reproduction techniques. Moreover, birthweight and EFW percentiles were markedly lower in the Galicia cohort, with significantly higher rates of neonates and fetuses below the 10th and 5th percentiles.

### 3.2. Model Development

Ultrasound alone demonstrated a limited predictive performance for identifying SGA neonates. When using an EFW < 10th percentile as the screening threshold, 302 of the 630 neonates born with a birthweight < 10th percentile were correctly identified, resulting in a DR of 47.9% (95% CI: 44.0 to 51.9). This corresponded to an SPR of 14.0%, with 546 of the 3992 fetuses having an EFW < 10th percentile, of which 244 were born with an adequate birthweight, resulting in an FPR of 7.3%. Similarly, for birthweight < 5th percentile, 176 of 364 neonates were correctly classified using an EFW < 5th percentile, yielding a DR of 48.4% (95% CI: 43.1 to 53.6) at an 8.5% SPR. Of the 338 fetuses with EFW < 5th percentile, 162 were born with an appropriate birthweight, resulting in an FPR of 4.5%.

Incorporation of maternal factors, biophysical, and biochemical markers led to modest but progressive improvements in model performance for the prediction of neonates born below the 10th and 5th birthweight percentiles (Table 3 and Table 4). The model based solely on maternal factors combined with EFW achieves an AUC of 0.851 (95% CI: 0.835 to 0.867), and the model using all available biomarkers achieves an AUC of 0.857 (95% CI: 0.842 to 0.873) for the prediction of neonates born < 10th percentile (Table 3). A similar pattern was observed for the prediction of birthweight < 5th percentile (Table 4): The AUC of the simplest model tested was 0.882 (95% CI: 0.864 to 0.900), and the AUC of the most complex model was 0.885 (95% CI: 0.868 to 0.903). Across all models, performance improved as additional variables were introduced, with only minimal differences between intermediate and full models. Fitted regression models with maternal demographic characteristics and medical history (maternal factors) and EFW, +/− MAP, +/− Doppler indices, +/− biochemical markers at 35 + 0 to 36 + 6 weeks’ gestation for prediction of SGA neonate with birthweight < 10th and <5th percentile are provided in Appendix A.

Calibration plots for each model are provided in Appendix A. The best calibration was obtained for the model using maternal factors combined with EFW, with a slope of 1.016 (95% CI: 0.914 to 1.123) and an intercept of 0.111 (95% CI: −0.068 to 0.294), although all models had adequate calibration results.

### 3.3. External Validation

To evaluate model performance in an independent population, external validation was conducted using data from Galicia. For predicting birthweight below the 10th percentile, the DR increased to 68.3%, 85.5%, 95.2%, and 96.2% at 10%, 20%, 30%, and 40% FPR, respectively. For birthweight below the 5th percentile, DR also increased to 74.7%, 89.3%, 98.7%, and 100% at the corresponding FPR thresholds. Model discrimination remained similar, with AUC values of 0.910 (95% CI: 0.886 to 0.934) for <10th percentile and 0.923 (95% CI: 0.902 to 0.944) for <5th percentile.

### 3.4. Performance of Screening for SGA Delivered Before 38 Weeks and Further Monitoring

When applying the predictive model developed in this study, DR for SGA neonates delivering < 38 weeks was considerably higher than when predicting for delivery at any gestational age, across all FPRs (Table 5, Figure 1).

Our results indicate that to achieve a high DR for SGA at birth, an SPR of approximately 40% would be required. In clinical terms, this would mean recommending a follow-up scan for all high-risk women within four weeks of the routine 35–36 weeks assessment. However, many of these pregnancies will have already reached delivery before that follow-up can occur. Table 6 shows the percentage of women identified as high risk in our study population who were still undelivered by 40 weeks of gestation, illustrating that the proportion of patients who would undergo a subsequent scan is considerably lower than the initial screen-positive group.

## 4. Discussion

### 4.1. Main Findings

The main findings of this study are as follows: First, that routine third-trimester assessment at 35–36 weeks in an unselected population can detect the majority of fetuses that will be born small, especially those delivering before 38 weeks; second, that although models based solely on ultrasound have only moderate predictive accuracy, the incorporation of maternal characteristics improves DR; third, that the addition of Doppler parameters and, to a lesser extent, angiogenic biomarkers such as PlGF and sFlt-1, provides only a marginal improvement in predictive performance, especially for the prediction of birthweight < 5th percentile; and fourth, that to achieve a DR of at least 80% for neonates with birthweight < 10th or <5th percentile, an SPR of approximately 40% is required, yet less than 40% of screen-positive women will remain undelivered by 40 weeks and therefore actually undergo further monitoring.

### 4.2. Comparison with Previous Studies

This study confirms and extends prior evidence that third-trimester ultrasonographic assessment at 35–36 weeks is a valuable tool for identifying fetuses at risk of being born SGA. Our findings demonstrate that a single routine scan at this gestational age can detect a substantial proportion of neonates with birthweight below the 10th or 5th percentiles. These results align with the conclusions of the ROTTUS trial, which reported that routine third-trimester ultrasound significantly improved detection of SGA compared to selective scanning based on symphysis-fundus height measurements, with a detection rate of 52.8% for <10th centile and 66.7% for <3rd centile neonates [30]. Third-trimester ultrasound performed systematically in low-risk pregnancies has already been shown to improve the detection rate of SGA and FGR fetuses compared to clinically guided ultrasound when other risk factors are present or when there is a clinical suspicion of fetal growth restriction [12,31]. In the POP study, universal third-trimester scan at 36 weeks in nulliparous women more than doubled the detection of SGA compared to clinically indicated ultrasonography [12].

However, the performance of ultrasound biometry alone was modest, in line with prior literature. For example, the recent studies by Adjahou et al., Ali et al., and Caradeux et al. reported detection rates around 50% for birthweight < 10th percentile at any gestational age using ultrasound alone [18,32,33], which is similar to the 47.9% found in our study. We have also demonstrated that the addition of maternal characteristics and, to a lesser extent, Doppler parameters, modestly improved predictive performance, which aligns with the studies from Cioabanu et al., Miranda et al., and Papastefanou et al., who demonstrated that multivariable models incorporating clinical and biophysical factors outperform ultrasound alone, particularly in detecting more severe growth restriction [19,34,35]. Ciobanu et al. in their 2019 study performed ultrasound in 35–37 weeks in 19,209 singleton pregnancies and, using a logistic regression analysis, obtained a DR of 75% (95% CI: 69 to 81; AUC 0.931, 95% CI: 0.914 to 0.945) at 10% with a combination of maternal factors and EFW and a detection rate of 80% (95% CI: 74 to 86; AUC 0.933, 95% CI: 0.917 to 0.949) by maternal factors, EFW, and UtAPI, MCA-PI, and PlGF MoM values [34]. Miranda et al. in their 2016 study, also using logistic regression analysis, obtained an AUC of 0.82 (95% CI: 0.77 to 0.85) for SGA detection with EFW alone, versus an AUC of 0.86 (95% CI: 0.83 to 0.89) for a model that included EFW, Doppler, maternal factors, and biomarkers (PlGF and estriol), and a low-birth-weight detection rate of 77% at a 10% false-positive rate [35]. Interestingly, their ultrasound scans were performed between 32 and 36 + 6 weeks of pregnancy, and it has been previously shown that the later assessment is associated with a better SGA prediction [11,18,33,36,37]. However, even with such combined approaches, achieving detection rates above 80% required an SPR of approximately 40% [38], which is consistent with our results. Finally, Papastefanou et al. evaluated a competing-risks model at 35–36 weeks’ gestation in a large cohort of 29,781 singleton pregnancies, incorporating EFW, maternal characteristics, obstetric history, and Doppler indices, achieving a DR of 65.8% (95% CI: 65.3 to 66.4) of all SGA neonates < 10th percentile at a 10% FPR, with a marginal improvement by adding UtA-PI and PlGF [19]. In our study, as in these previously discussed studies, the incremental gain in detection with biomarkers was also minimal, particularly when compared to the addition of maternal and Doppler data, and may not justify the increased complexity and cost in low-risk settings. Finally, and consistent with these previous studies, we also found better performance for the prediction of SGA neonates that were delivered <38 weeks, even if a specific model was not fitted for that purpose.

Importantly, we demonstrated that only a minority of screen-positive pregnancies remained undelivered by 40 weeks. This has major implications for clinical strategies based on follow-up ultrasound at term, as many women will have already delivered before a second evaluation can occur, therefore reducing unnecessary anxiety, overuse of resources, and diminishing returns. This dilution of clinical impact despite high theoretical performance was also highlighted in the work of van Roekel et al., who cautioned against over-reliance on false-positive findings in low-risk populations, as they may not lead to meaningful interventions [39].

Overall, our findings support the continued use of third-trimester screening in universal or selected populations but underscore the need to optimize timing, streamline risk stratification, and tailor follow-up protocols to ensure the true clinical benefit of SGA detection is realized.

### 4.3. Strengths and Limitations

A major strength of this study is its large, multicenter design, which included an unselected population across several regions, enhancing the generalizability of the findings. The study used prospectively collected data and adhered to a standardized ultrasound protocol at 35–36 weeks, closely reflecting real-world clinical practice. Importantly, external validation was performed in an independent cohort, strengthening the robustness and reproducibility of the models. The inclusion of different combinations of maternal characteristics, biometry, Doppler parameters, and biochemical markers also allowed for a comprehensive comparison of predictive strategies, offering practical insight into their incremental value and limitations.

Nevertheless, the study has several limitations. First, although external validation was conducted, performance varied substantially between populations, which may reflect differences in maternal characteristics, clinical practices, or healthcare settings. Furthermore, the cohort used for external validation significantly differed from the original cohort, likely due to a selection bias in a tertiary referral center where many low-risk women opt to deliver in smaller private centers. Since this is a higher-risk population, the improved results in this cohort are expected and consistent. Second, biomarkers were not available in all participants, which limited their inclusion to a subset and may have reduced power for fully assessing their contribution. Third, although we report detection rates at various screen-positive rates, we did not assess the clinical outcomes associated with detection, such as reductions in perinatal morbidity or mortality. Finally, the models were designed for prediction at a single time point (35–36 weeks), and we did not explore dynamic models or longitudinal changes, which might further improve accuracy.

### 4.4. Clinical Implications

Given the results of our study, the implementation of third-trimester ultrasound screening at 35–36 weeks, combined with maternal factors, has the potential to identify the majority of neonates with birthweight below the 10th and 5th percentiles and should therefore be considered as part of routine pregnancy surveillance. However, our findings also demonstrate that predictive performance is highly dependent on the SPR and the timing of delivery, which has important consequences for the design and effectiveness of follow-up strategies.

While the addition of maternal factors and MAP may improve DR compared to ultrasound alone, the additional benefit from including other biomarkers, such as Doppler indices, and more so, PlGF and sFlt-1, is limited and may not justify their broader use in universal screening protocols due to added complexity and cost [40]. To achieve DR above 80%, an SPR of 40% or higher is required; however, only a minority of screen-positive women, less than 40%, will remain undelivered at 40 weeks. This means that although a large proportion of the population would initially qualify for increased surveillance, many will deliver before a follow-up scan is due, thereby substantially reducing the actual number of additional scans required. This limits the burden on healthcare resources while still enabling improved detection.

Overall, our findings support the use of third-trimester screening in universal or unselected populations but underscore the need to optimize timing, streamline risk stratification, and tailor follow-up protocols to ensure the true clinical benefit of SGA detection is realized. Future research should explore adaptive follow-up pathways based on individualized risk and time-to-delivery, as well as the potential role of earlier assessments or repeated measurements to refine predictions and guide interventions. Additionally, outcome-based studies are needed to assess whether improved detection translates into meaningful perinatal benefit and to identify the thresholds at which intervention does more good than harm.

## 5. Conclusions

Routine third-trimester assessment at 35–36 weeks in an unselected population can identify the majority of fetuses that will be born small. While ultrasound-based screening alone has moderate predictive performance, DR may improve with the addition of maternal risk factors, Doppler parameters, and, to a lesser extent, biochemical markers. Importantly, achieving high detection rates requires broad screening thresholds, yet a substantial proportion of screen-positive women will deliver before any follow-up can occur, limiting the practical impact of such strategies.

These findings highlight the need for more refined and timely approaches to optimize the clinical utility of third-trimester fetal growth assessment.

## Figures and Tables

**Figure 1 medicina-61-01626-f001:**
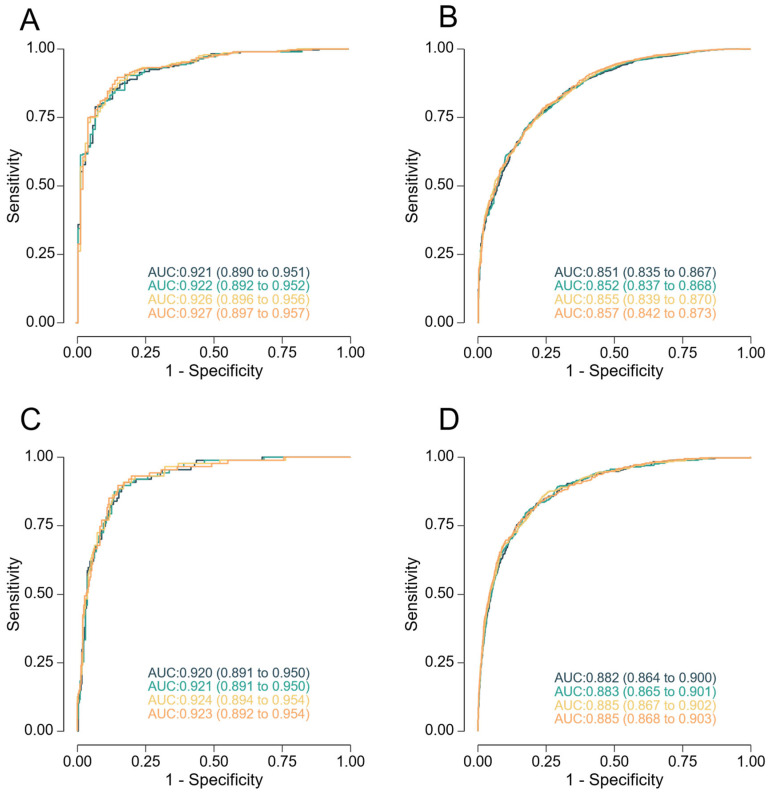
Receiver-operating characteristics curves for screening by the different models at 35 + 0 to 36 + 6 weeks of gestation for prediction of neonates with a birthweight < 10th percentile (panels (**A**,**B**)) and <5th (panels (**C**,**D**)), delivering < 38 weeks (panels (**A**,**C**)), and at any gestational age (panels (**B**,**D**)). The model combining maternal factors with estimated fetal weight is represented in black, addition of mean arterial pressure in green, addition of Doppler indices in yellow, and addition of PlGF and sFlt-1 in orange.

**Table 1 medicina-61-01626-t001:** Maternal and pregnancy characteristics of the study population according to SGA status.

	Non-SGA (N = 3727)	SGA (N = 816)	*p*-Value
Maternal age (years)	34.0 (30.0; 37.0)	34.0 (30.0; 37.0)	0.01
Maternal height	163 (160; 168)	162 (158; 165)	<0.001
Maternal weight	75.6 (68.0; 85.0)	69.3 (61.8; 78.3)	<0.001
Smokers	372 (10.0%)	154 (18.9%)	<0.001
Spontaneous conception	3511 (94.2%)	763 (93.5%)	0.461
Systemic lupus erythematosus	6 (0.2%)	5 (0.6%)	0.033
Antiphospholipid syndrome	7 (0.2%)	4 (0.5%)	0.119
Diabetes mellitus	26 (0.7%)	1 (0.1%)	0.073
Chronic hypertension	37 (1.0%)	12 (1.5%)	0.259
Family history of PE	149 (4.0%)	26 (3.2%)	0.315
Current diagnosis of:			
Gestational diabetes mellitus	153 (4.1%)	35 (4.3%)	0.772
PE	17 (0.5%)	15 (1.8%)	<0.001
Pregnancy-induced hypertension	28 (0.8%)	21 (2.6%)	<0.001
Personal obstetric history:			
Nulliparous	1701 (45.6%)	508 (62.3%)	<0.001
Parous—no FGR—no PE	1856 (49.8%)	231 (28.3%)	
Parous—no FGR—PE	38 (1.0%)	6 (0.7%)	
Parous—FGR—no PE	120 (3.2%)	67 (8.2%)	
Parous—FGR—PE	12 (0.3%)	4 (0.5%)	
Gestational age at ultrasound (weeks)	35.6 (35.3; 36.1)	35.7 (35.3; 36.1)	0.007
EFW (grams)	2720 (2540; 2910)	2380 (2220; 2550)	<0.001
EFW percentile	55.5 (27.9; 79.2)	8.56 (1.68; 24.5)	<0.001
EFW z-score	0.14 (−0.59; 0.81)	−1.37 (−2.13; −0.69)	<0.001
MAP MoMs	0.98 (0.92; 1.03)	0.99 (0.94; 1.06)	<0.001
UA-PI	0.89 (0.78; 1.00)	0.94 (0.83; 1.06)	<0.001
MCA-PI	1.67 (1.45; 1.92)	1.60 (1.40; 1.86)	<0.001
UtA-PI MoMs	0.96 (0.82; 1.12)	1.02 (0.86; 1.25)	<0.001
sFlt-1 MoMs	0.98 (0.70; 1.37)	1.05 (0.72; 1.66)	<0.001
PlGF MoMs	0.99 (0.53; 1.78)	0.61 (0.34; 1.24)	<0.001
Gestational age at delivery (weeks)	39.9 (39.0; 40.7)	39.3 (38.1; 40.3)	<0.001
Birthweight (grams)	3380 (3150; 3640)	2680 (2460; 2840)	<0.001
Birthweight percentile	46.1 (26.6; 69.2)	3.43 (1.32; 6.60)	<0.001

Categorical variables are described as n (%) and continuous variables as median (Q1; Q3). SGA: small for gestational age; PE: preeclampsia; FGR: fetal growth restriction; EFW: estimated fetal weight; MAP: mean arterial pressure; UA-PI: umbilical artery pulsatility index; MCA-PI: middle cerebral artery pulsatility index; UtA-PI: uterine artery pulsatility index; PlGF: placental growth factor; sFlt-1: soluble fms-like tyrosine kinase-1.

**Table 2 medicina-61-01626-t002:** Maternal and pregnancy characteristics in the development (Murcia and Madrid) and validation (Galicia) cohorts.

	Development Cohort	Validation Cohort	*p*-Value
	(N = 3992)	(N = 551)	
Maternal age (years)	33.0 (30.0; 37.0)	36.0 (32.0; 39.0)	<0.001
Maternal height	163 (159; 167)	162 (158; 166)	<0.001
Maternal weight	75.5 (68.2; 84.5)	65.0 (58.0; 75.6)	<0.001
Smokers	448 (11.2%)	78 (14.2%)	0.047
Spontaneous conception	3804 (95.3%)	470 (85.3%)	<0.001
Systemic lupus erythematosus	10 (0.3%)	1 (0.2%)	1
Antiphospholipid syndrome	10 (0.3%)	1 (0.2%)	1
Diabetes mellitus	16 (0.4%)	11 (2.0%)	<0.001
Chronic hypertension	26 (0.7%)	23 (4.2%)	<0.001
Family history of PE	166 (4.2%)	9 (1.6%)	0.003
Current diagnosis of:			
Gestational diabetes mellitus	161 (4.0%)	27 (4.9%)	0.360
PE	18 (0.5%)	14 (2.5%)	<0.001
Pregnancy-induced hypertension	23 (0.6%)	26 (4.7%)	<0.001
Personal obstetric history:			
Nulliparous	1875 (47.0%)	334 (60.6%)	<0.001
Parous—no FGR—no PE	1909 (47.8%)	178 (32.3%)	
Parous—no FGR—PE	38 (1.0%)	6 (1.1%)	
Parous—FGR—no PE	159 (4.0%)	28 (5.1%)	
Parous—FGR—PE	11 (0.3%)	5 (0.9%)	
Gestational age at ultrasound (weeks)	35.6 (35.3; 36.1)	35.7 (35.3; 36.1)	<0.001
EFW (grams)	2670 (2478; 2867)	2620 (2357; 2881)	0.001
EFW percentile	47.5 (20.7; 74.6)	37.2 (4.81; 75.5)	<0.001
EFW z-score	−0.06 (−0.82; 0.66)	−0.33 (−1.66; 0.69)	<0.001
Fetuses EFW < 10th percentile	546 (13.7%)	166 (30.1%)	<0.001
Fetuses EFW < 5th percentile	338 (8.5%)	140 (25.4%)	<0.001
Gestational age at delivery (weeks)	39.9 (38.9; 40.7)	39.1 (38.2, 40.0)	<0.001
Birthweight (grams)	3290 (3010; 3580)	3080 (2675; 3420)	<0.001
Birthweight percentile	39.0 (16.8; 64.8)	24.2 (3.81; 54.2)	<0.001
Neonates born < 10th percentile	630 (15.8%)	186 (33.8%)	<0.001
Neonates born < 5th percentile	364 (9.1%)	150 (27.2%)	<0.001

Categorical variables are described as n (%) and continuous variables as median (Q1; Q3). PE: preeclampsia; FGR: fetal growth restriction; EFW: estimated fetal weight.

**Table 3 medicina-61-01626-t003:** Area under the ROC curve, detection rates, and screen-positive rates with 95% confidence intervals for each algorithm in the prediction of birthweight centile < 10% after the 35–36 weeks assessment, at different fixed false positive rates.

Model		At 10% FPR	At 20% FPR	At 30% FPR	At 40% FPR	AUC (95% CI)
Maternal factors + EFW	DR (95% CI)	57.0 (53.0 to 60.9)	72.2 (68.6 to 75.7)	83.0 (79.9 to 85.9)	88.4 (85.7 to 90.8)	0.851 (0.835 to 0.867)
	SPR (%)	17.4	28.2	38.4	47.6	
Maternal factors + EFW + MAP	DR (95% CI)	58.1 (54.1 to 62.0)	71.9 (68.2 to 75.4)	83.0 (79.9 to 85.9)	90.0 (87.4 to 92.2)	0.852 (0.837 to 0.868)
	SPR (%)	17.6	28.2	38.4	47.9	
Maternal factors + EFW + MAP + UA-PI + MCA-PI + UtA-PI	DR (95% CI)	58.4 (54.5 to 62.3)	72.7 (69.0 to 76.1)	82.7 (79.5 to 85.6)	89.1 (86.3 to 91.4)	0.855 (0.839 to 0.870)
	SPR (%)	17.6	28.3	38.3	47.7	
Maternal factors + EFW + MAP + UA-PI + MCA-PI + UtA-PI + PlGF + sFlt-1	DR (95% CI)	60.2 (56.2 to 64.0)	73.3 (69.7 to 76.8)	82.7 (79.5 to 85.6)	89.1 (86.3 to 91.4)	0.857 (0.842 to 0.873)
	SPR (%)	17.9	28.4	38.3	47.7	

FPR: false positive rate; AUC: area under the ROC curve; CI: confidence interval; DR: detection rate; SPR: screen-positive rate; EFW: estimated fetal weight; MAP: mean arterial pressure; UA-PI: umbilical artery pulsatility index; MCA-PI: middle cerebral artery pulsatility index; UtA-PI: uterine artery pulsatility index; PlGF: placental growth factor; sFlt-1: soluble fms-like tyrosine kinase-1. EFW was used as a z-score and biomarkers as MoMs.

**Table 4 medicina-61-01626-t004:** Area under the ROC curve, detection rates, and screen-positive rates with 95% confidence intervals for each algorithm in the prediction of birthweight centile < 5% after the 35–36 weeks assessment, at different fixed false positive rates.

Model		At 10% FPR	At 20% FPR	At 30% FPR	At 40% FPR	AUC (95% CI)
Maternal factors + EFW	DR (95% CI)	65.7 (60.5 to 70.5)	80.5 (76.0 to 84.4)	88.2 (84.4 to 91.3)	92.3 (89.1 to 94.8)	0.882 (0.864 to 0.900)
	SPR	15.1	25.5	35.3	44.7	
Maternal factors + EFW + MAP	DR (95% CI)	67.0 (61.9 to 71.8)	81.3 (76.9 to 85.2)	89.3 (85.7 to 92.3)	92.3 (89.1 to 94.8)	0.883 (0.865 to 0.901)
	SPR	15.2	25.6	35.4	44.7	
Maternal factors + EFW + MAP + UA-PI + MCA-PI + UtA-PI	DR (95% CI)	68.4 (63.4 to 73.2)	80.8 (76.3 to 84.7)	88.4 (84.7 to 91.6)	92.9 (89.7 to 95.3)	0.885 (0.867 to 0.902)
	SPR	15.3	25.5	35.3	44.8	
Maternal factors + EFW + MAP + UA-PI + MCA-PI + UtA-PI + PlGF + sFlt-1	DR (95% CI)	69.2 (64.2 to 73.9)	80.8 (76.3 to 84.7)	87.4 (83.5 to 90.6)	91.5 (88.1 to 94.1)	0.885 (0.868 to 0.903)
	SPR	15.4	25.5	35.2	44.7	

FPR: false positive rate; AUC: area under the ROC curve; CI: confidence interval; DR: detection rate; SPR: screen-positive rate; EFW: estimated fetal weight; MAP: mean arterial pressure; UA-PI: umbilical artery pulsatility index; MCA-PI: middle cerebral artery pulsatility index; UtA-PI: uterine artery pulsatility index; PlGF: placental growth factor; sFlt-1: soluble fms-like tyrosine kinase-1. EFW was used as a z-score and biomarkers as MoMs.

**Table 5 medicina-61-01626-t005:** Area under the ROC curve, detection rates, and screen positive rates with 95% confidence intervals for each algorithm in the prediction of birthweight percentile < 10th < 5th when delivery occurred <38 weeks of gestation, at different fixed false positive rates.

Model						
Prediction of BW < 10th		10% FPR	At 20% FPR	At 30% FPR	At 40% FPR	AUC (95% CI)
Maternal factors + EFW	DR (95% CI)	77.5 (68.6 to 84.9)	90.1 (83.0 to 95.0)	93.7 (87.4 to 97.4)	96.4 (91.0 to 99.0)	0.921 (0.87 to 0.96)
	SPR	29.0	39.7	47.9	55.9	
Maternal factors + EFW + MAP	DR (95% CI)	82.0 (73.6 to 88.6)	90.1 (83.0 to 95.0)	93.7 (87.4 to 97.4)	98.2 (93.6 to 99.8)	0.922 (0.88 to 0.96)
	SPR	30.3	39.7	47.9	56.4	
Maternal factors + EFW + MAP + UA-PI + MCA-PI + UtA-PI	DR (95% CI)	81.1 (72.6 to 87.9)	89.2 (81.9 to 94.3)	95.5 (89.8 to 98.5)	96.4 (91.0 to 99.0)	0.926 (0.90 to 0.96)
	SPR	30.0	39.5	48.5	55.9	
Maternal factors + EFW + MAP + UA-PI + MCA-PI + UtA-PI + PlGF + sFlt-1	DR (95% CI)	82.0 (73.6 to 88.6)	91.0 (84.1 to 95.6)	95.5 (89.8 to 98.5)	97.3 (92.3 to 99.4)	0.927 (0.89 to 0.96)
	SPR	30.3	40.0	48.5	56.2	
Prediction of BW < 5th		10% FPR	At 20% FPR	At 30% FPR	At 40% FPR	AUC (95% CI)
Maternal factors + EFW	DR (95% CI)	74.7 (64.3 to 83.4)	90.8 (82.7 to 96.0)	93.1 (85.6 to 97.4)	95.4 (88.6 to 98.7)	0.92 (0.891 to 0.95)
	SPR	24.1	35.6	43.8	52.1	
Maternal factors + EFW + MAP	DR (95% CI)	75.9 (65.5 to 84.4)	90.8 (82.7 to 96.0)	94.3 (8.1 to 98.1)	97.7 (91.9 to 99.7)	0.921 (0.891 to 0.95)
	SPR	24.4	35.6	44.1	52.6	
Maternal factors + EFW + MAP + UA-PI + MCA-PI + UtA-PI	DR (95% CI)	73.6 (63.0 to 82.5)	92.0 (84.1 to 96.7)	93.1 (85.6 to 97.4)	97.7 (91.9 to 99.7)	0.924 (0.894 to 0.954)
	SPR	23.8	35.9	43.8	52.6	
Maternal factors + EFW + MAP + UA-PI + MCA-PI + UtA-PI + PlGF + sFlt-1	DR (95% CI)	77.01 (66.8 to 85.4)	93.1 (85.6 to 97.4)	94.3 (87.1 to 98.1)	96.6 (90.3 to 99.3)	0.923 (0.892 to 0.954)
	SPR	24.6	36.2	44.1	52.3	

FPR: false positive rate; AUC: area under the ROC curve; CI: confidence interval; DR: detection rate; SPR: screen-positive rate; EFW: estimated fetal weight; MAP: mean arterial pressure; UA-PI: umbilical artery pulsatility index; MCA-PI: middle cerebral artery pulsatility index; UtA-PI: uterine artery pulsatility index; PlGF: placental growth factor; sFlt-1: soluble fms-like tyrosine kinase-1. EFW was used as a z-score and biomarkers as MoMs.

**Table 6 medicina-61-01626-t006:** Percentage of women classified as high risk who had not delivered at 40 weeks of gestational age according to the model and screen-positive rate used.

Model	Percentage (%) of Women Undelivered at 40 Weeks			
Prediction BW < 10th	If FPR 10%	If FPR 20%	If FPR 30%	If FPR 40%
Maternal factors + EFW	35.16	38.45	39.19	40.45
Maternal factors + EFW + MAP	32.95	37.37	38.60	40.19
Maternal factors + EFW + MAP + UA-PI/+ MCA-PI/+ UtA-PI	33.14	37.02	38.98	39.42
Maternal factors + EFW + MAP + UA-PI/+ MCA-PI/+ UtA-PI + PlGF/sFlt-1	30.81	35.30	37.74	38.32
Prediction BW < 5th				
Maternal factors + EFW	33.11	37.62	39.42	39.87
Maternal factors + EFW + MAP	32.51	37.41	38.60	39.25
Maternal factors + EFW + MAP + UA-PI/+ MCA-PI/+ UtA-PI	30.77	36.11	38.89	39.15
Maternal factors + EFW + MAP + UA-PI/+ MCA-PI/+ UtA-PI + PlGF/sFlt-1	28.66	34.35	36.87	37.97

FPR: false positive rate; EFW: estimated fetal weight; MAP: mean arterial pressure; UA-PI: umbilical artery pulsatility index; MCA-PI: middle cerebral artery pulsatility index; UtA-PI: uterine artery pulsatility index; PlGF: placental growth factor; sFlt-1: soluble fms-like tyrosine kinase-1.

## Data Availability

The raw data supporting the conclusions of this article will be made available by the authors on request.

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
