# Peer review of "Prediction of Small for Gestational Age and Growth-Restricted Neonates at 35 to 36 Weeks of Gestation: A Multicenter Cohort Study"

_medicina, 2025, doi:10.3390/medicina61091626_

Round 1

Reviewer 1 Report

Comments and Suggestions for Authors The authors present a manuscript which aims to set up a model for identifying the small-for-gestational age and growth restricted neonates with a gestational age of 35 to 36 weeks. The research aims to investigate the potential benefits of adopting sonographic and biochemical markers on predicting the delivery of small-for-gestational age neonates and newborns with growth restriction. Predicting the delivery of small-for-gestational age and growth restricted newborns is a significant topic which is frequently encountered in daily clinical practice. This topic also addresses a specific gap in the fields of obstetrics and perinatology as it targets the improvement in the well-being of fetuses with biometrical features that are not compatible with gestational age. There are a number of studies which focus on the detection of growth restricted and small-for-gestational age neonates by either sonographic or biochemical markers. However, this study has additional merit as it aims to establish a model by combining sonographic and biochemical measurements as well as Doppler ultrasonography parameters. The methodology of the study has been described proficiently and the recruitment of development and validation cohorts has been well explained. The conclusions achieved by this study comply with the evidence and arguments presented and these conclusions address the main question posed. Moreover, parts of the discussion have been separately dedicated to an extensive review of related literature and detailed examination of strengths, limitations and possible clinical implications of the authors' findings. Additionally, it should be noted that all the references are appropriate, relevant and up-to-date and all tables are required and have adequate quality.

Author Response

Reviewer 1

The authors present a manuscript which aims to set up a model for identifying the small-for-gestational age and growth restricted neonates with a gestational age of 35 to 36 weeks. The research aims to investigate the potential benefits of adopting sonographic and biochemical markers on predicting the delivery of small-for-gestational age neonates and newborns with growth restriction. Predicting the delivery of small-for-gestational age and growth restricted newborns is a significant topic which is frequently encountered in daily clinical practice. This topic also addresses a specific gap in the fields of obstetrics and perinatology as it targets the improvement in the well-being of fetuses with biometrical features that are not compatible with gestational age. There are a number of studies which focus on the detection of growth restricted and small-for-gestational age neonates by either sonographic or biochemical markers. However, this study has additional merit as it aims to establish a model by combining sonographic and biochemical measurements as well as Doppler ultrasonography parameters. The methodology of the study has been described proficiently and the recruitment of development and validation cohorts has been well explained. The conclusions achieved by this study comply with the evidence and arguments presented and these conclusions address the main question posed. Moreover, parts of the discussion have been separately dedicated to an extensive review of related literature and detailed examination of strengths, limitations and possible clinical implications of the authors' findings. Additionally, it should be noted that all the references are appropriate, relevant and up-to-date and all tables are required and have adequate quality.

Thank you very much for your kind comments.

Reviewer 2 Report

Comments and Suggestions for Authors

I would like to thank the Authors and the Editor for the opportunity, having read with interest the article.

The manuscript by Martin Alonso et al is s retrospective cohort study in three Spanish centers aimed to assess the performance of combined maternal factors and biomarkers, including ultrasound estimated fetal weight (EFW), Doppler indices, mean arterial pressure (MAP), and angiogenic biomarkers, for predicting SGA neonates after a routine 35–36 week scan in an unselected population. While the article presents interesting insights, I would suggest some considerations before publication.

Although the issue could be potentially interesting, several major criticisms have to be raised and some revisions are required in order to improve the quality of the manuscript:  in particular

General point

A careful revision of literature should be considered important to improve the quality of the paper.

A wide revision of English language and punctuation should be considered mandatory to improve the quality of the paper.

The methods section seems to be confusing and not accurate.

The reference section should be checked and corrected and added according to the authors guidelines.

Besides these major concerns, other issue have to be raised and some typing mistakes exists. I have some comments.

Specific points:

  • INTRODUCTION

The introduction section seems to be not accurate. Please try to reach out clinical features.

  • METHODS

Which is the argument for choosing January 2017 as starting year?

The methodology is essential and needs to be described in more proper detail. All the above must be made absolutely clear and correct.

  • Results

Page 10 line 278 “an independent popula-tion,” please correct.

Line 283 “sim-ilar” please correct

Line 293 “al-ready” please correct

  • Discussion

Tabacco et al in 2023 (PMID: 37623210) well described in a review of the literature the specific etiology and molecular mechanisms of pre-eclampsia that are still poorly known and could have a variety of causes, such as altered angiogenesis, inflammations, maternal infections, obesity, metabolic disorders. One of the most promising areas under investigation is the maternal angiogenic factor imbalance and its effects on vascular function; methods of risk stratification using ratios of proangiogenic factors—such as PlGF—and antiangiogenic factors—such as sFLT1 and sENG—have high detection rates for preterm pre-eclampsia. A comment on it would be appreciate.

5) TABLES

- Try to find a better and easier formatting for the tables. 

- Use the same character for each Table. Please correct. 

6) REFERENCE

- Add the suggested references in the right place. Please modify.

Comments on the Quality of English Language

see comment

Author Response

Reviewer 2

I would like to thank the Authors and the Editor for the opportunity, having read with interest the article.

The manuscript by Martin Alonso et al is s retrospective cohort study in three Spanish centers aimed to assess the performance of combined maternal factors and biomarkers, including ultrasound estimated fetal weight (EFW), Doppler indices, mean arterial pressure (MAP), and angiogenic biomarkers, for predicting SGA neonates after a routine 35–36 week scan in an unselected population. While the article presents interesting insights, I would suggest some considerations before publication.

Although the issue could be potentially interesting, several major criticisms have to be raised and some revisions are required in order to improve the quality of the manuscript:  in particular

General point

A careful revision of literature should be considered important to improve the quality of the paper.

Thank you for this observation. We have undertaken a thorough review of the existing literature in the preparation of this manuscript and aimed to include the most relevant and methodologically robust studies in our discussion. However, we appreciate that some important references may have been inadvertently missed. If there are specific studies you believe should be added or revised, we would be grateful if you could kindly indicate them so we can address this accordingly.

A wide revision of English language and punctuation should be considered mandatory to improve the quality of the paper.

Thank you for your suggestion. The manuscript was revised for language and clarity with the support of our university's English editing service (native speakers). In addition, one of our co-authors, Alan Wright, M.P.H, who is a native English speaker, carefully reviewed the manuscript and suggested only a few modifications throughout the manuscript.

The methods section seems to be confusing and not accurate.

Thank you for this observation. To improve clarity and readability, we have organized the Materials and Methods section into three distinct subsections: 2.1. Study Design and Population, 2.2. Outcome Measures, and 2.3. Statistical Analysis. As we have received somewhat contradictory feedback from different reviewers regarding this section, and given that the methodological score is overall very good, we would be grateful for more specific guidance on which aspects are considered confusing or inaccurate, so we can address them appropriately.

The reference section should be checked and corrected and added according to the authors guidelines.

Thank you, done.

Besides these major concerns, other issue have to be raised and some typing mistakes exists. I have some comments.

Specific points:

  • INTRODUCTION

The introduction section seems to be not accurate. Please try to reach out clinical features.

Thank you for your comment. The introduction follows a standard structure, highlighting the clinical relevance of SGA, reviewing current evidence, and stating our objectives. If specific inaccuracies or missing clinical points can be identified, we would be happy to address them.

  • METHODS

Which is the argument for choosing January 2017 as starting year?

Thank you for your comment. Although several logistical reasons contributed to choosing January 2017, the main rationale, stated in the manuscript, is that this cohort was part of a previous prospective clinical study (ref. 23), ensuring prospectively collected, high-quality data despite the retrospective design.

  • Results

Page 10 line 278 “an independent popula-tion,” please correct.

Line 283 “sim-ilar” please correct

Line 293 “al-ready” please correct

Thank you very much, done.

  • Discussion

Tabacco et al in 2023 (PMID: 37623210) well described in a review of the literature the specific etiology and molecular mechanisms of pre-eclampsia that are still poorly known and could have a variety of causes, such as altered angiogenesis, inflammations, maternal infections, obesity, metabolic disorders. One of the most promising areas under investigation is the maternal angiogenic factor imbalance and its effects on vascular function; methods of risk stratification using ratios of proangiogenic factors—such as PlGF—and antiangiogenic factors—such as sFLT1 and sENG—have high detection rates for preterm pre-eclampsia. A comment on it would be appreciate.

Thank you for the suggestion. While we acknowledge the excellent review by Tabacco et al., our study focuses specifically on the prediction of fetal smallness. Although preeclampsia and SGA often coexist, they are distinct conditions, and most SGA cases occur without maternal preeclampsia. The role of angiogenic markers in identifying SGA has been addressed in our study (refs. 19, 33, 34). A detailed discussion on preeclampsia falls outside the scope of this paper but will be considered in future work.

5) TABLES

- Try to find a better and easier formatting for the tables.

Thank you for your comment. Tables 1 and 2 are descriptive and follow the journal’s standard format. Tables 3 and 4 present the key performance results (detection and screen-positive rates at different false positive rates, and areas under the curves), which we believe are clinically relevant and consistent with previous studies. If there are specific formatting issues you’d like us to address, we would appreciate further guidance.

- Use the same character for each Table. Please correct. 

Thank you, reviewed. All tables are in Palatino linot 10.

6) REFERENCE

- Add the suggested references in the right place. Please modify.

See comment above. Thank you.

Reviewer 3 Report

Comments and Suggestions for Authors

The manuscript addresses an important clinical challenge: optimizing third-trimester screening for small-for-gestational-age (SGA) and growth-restricted (FGR) fetuses. The authors provide a large, multicenter dataset and test multiple predictive models with external validation, which strengthens the reliability of findings. The study is timely, methodologically solid, and clinically relevant. However, some aspects of clarity, clinical interpretation, and methodological reporting require improvement.

  • Clinical implications need clearer framing.

    • The study shows that achieving >80% detection requires ~40% screen-positive rate, yet less than 40% of screen-positive women remain undelivered at 40 weeks. This paradox should be discussed more thoroughly in terms of real-world impact (e.g., unnecessary anxiety, overuse of resources, and diminishing returns).

  • Selection bias in external validation.

    • The Galicia cohort differed significantly (older, smaller, higher comorbidities). The authors should clarify how this affects model generalizability and whether re-calibration is needed for different populations.

  • Cost-effectiveness of biomarkers.

    • Given the minimal gain from PlGF and sFlt-1, the discussion should explicitly address whether their inclusion is justifiable in universal screening programs. We recommend citing and discussing https://pubmed.ncbi.nlm.nih.gov/37539675/

  • Outcome relevance.

    • The study focuses on prediction of SGA/FGR, but does not link improved detection to perinatal outcomes. Without outcome data, it remains uncertain whether increased detection translates into clinical benefit.

Author Response

Reviewer 3

The manuscript addresses an important clinical challenge: optimizing third-trimester screening for small-for-gestational-age (SGA) and growth-restricted (FGR) fetuses. The authors provide a large, multicenter dataset and test multiple predictive models with external validation, which strengthens the reliability of findings. The study is timely, methodologically solid, and clinically relevant. However, some aspects of clarity, clinical interpretation, and methodological reporting require improvement.

  • Clinical implications need clearer framing.
    • The study shows that achieving >80% detection requires ~40% screen-positive rate, yet less than 40% of screen-positive women remain undelivered at 40 weeks. This paradox should be discussed more thoroughly in terms of real-world impact (e.g., unnecessary anxiety, overuse of resources, and diminishing returns).

Thank you for your comment. We agree that this is a key finding and have emphasized it further in the discussion:

 “Importantly, we demonstrated that only a minority of screen-positive pregnancies remained undelivered by 40 weeks. This has major implications for clinical strategies based on follow-up ultrasound at term, as many women will have already delivered before a second evaluation can occur, therefore reducing unnecessary anxiety, overuse of resources, and diminishing returns.”

  • Selection bias in external validation.
    • The Galicia cohort differed significantly (older, smaller, higher comorbidities). The authors should clarify how this affects model generalizability and whether re-calibration is needed for different populations.

Thank you for your comment. We have added the folloeing in the discussion:

“Besides, the cohort used for external validation significantly differed from the original cohort, likely due to a selection bias in a tertiary referral center where many low-risk women opt to deliver in smaller private centers. Since this is a higher risk population, the improved results in this cohort are expected and consistent.”

  • Cost-effectiveness of biomarkers.
    • Given the minimal gain from PlGF and sFlt-1, the discussion should explicitly address whether their inclusion is justifiable in universal screening programs. We recommend citing and discussing https://pubmed.ncbi.nlm.nih.gov/37539675/

Thank you, we have added the reference after the following statement: “the additional benefit from including other biomarkers such as Doppler indices and more so, PlGF and sFlt-1 is limited and may not justify their broader use in universal screening protocols due to added complexity and cost”

  • Outcome relevance.
    • The study focuses on prediction of SGA/FGR, but does not link improved detection to perinatal outcomes. Without outcome data, it remains uncertain whether increased detection translates into clinical benefit.

Thank you very much for your comment, which we fully share. This has been included in the study limitations section (pag 16: “Third, although we report detection rates at various screen-positive rates, we did not assess the clinical outcomes associated with detection, such as reductions in perinatal morbidity or mortality. ” ), and in the clinical implications section (pag 17: “Overall, our findings support the use of third-trimester screening in universal or unselected populations but underscore the need to optimize timing, streamline risk stratification, and tailor follow-up protocols to ensure the true clinical benefit of SGA detection is realized. Future research should explore adaptive follow-up pathways based on individualized risk and time-to-delivery, as well as the potential role of earlier assessments or repeated measurements to refine predictions and guide interventions. Additionally, outcome-based studies are needed to assess whether improved detection translates into meaningful perinatal benefit and to identify the thresholds at which intervention does more good than harm.”) as a proposal for future work.

Round 2

Reviewer 2 Report

Comments and Suggestions for Authors

I would like to thank the Authors and the Editor for the opportunity, having read with interest the reviewed article by the authors.

Despite the Authors have improved their manuscript, I still do have serious concerns about the quality and the importance of the clinical messages

  • Discussion

Tabacco et al in 2023 (PMID: 37623210) well described in a review of the literature the specific etiology and molecular mechanisms of pre-eclampsia that are still poorly known and could have a variety of causes, such as altered angiogenesis, inflammations, maternal infections, obesity, metabolic disorders. One of the most promising areas under investigation is the maternal angiogenic factor imbalance and its effects on vascular function; methods of risk stratification using ratios of proangiogenic factors—such as PlGF—and antiangiogenic factors—such as sFLT1 and sENG—have high detection rates for preterm pre-eclampsia. A comment on it would be appreciate.

Author Response

Despite the Authors have improved their manuscript, I still do have serious concerns about the quality and the importance of the clinical messages

Discussion

Tabacco et al in 2023 (PMID: 37623210) well described in a review of the literature the specific etiology and molecular mechanisms of pre-eclampsia that are still poorly known and could have a variety of causes, such as altered angiogenesis, inflammations, maternal infections, obesity, metabolic disorders. One of the most promising areas under investigation is the maternal angiogenic factor imbalance and its effects on vascular function; methods of risk stratification using ratios of proangiogenic factors—such as PlGF—and antiangiogenic factors—such as sFLT1 and sENG—have high detection rates for preterm pre-eclampsia. A comment on it would be appreciate.

We appreciate your positive comment on our article and your suggestion to add the proposed citation. However, this article focuses on diagnosing small fetuses, and in line with other existing articles on the subject (ref 19, 33 and 34), the predictive role of angiogenic markers is limited. Furthermore, while preeclampsia and growth restriction are interrelated pathologies in terms of their pathophysiology and presentation, this study does not consider mothers diagnosed with preeclampsia, nor does the article mention the origin of this disease. Nevertheless, we have included the proposed citation after the following statement: and serum concentrations of PlGF and sFlt-1.